# Complementary antibody lineages achieve neutralization breadth in an HIV-1 infected elite neutralizer

**Jelle van Schooten**[1], **Anna Schorcht**[1], **Elinaz Farokhi**[2], **Jeffrey C. Umotoy**[1], **Hongmei Gao**[3], **Tom L. G. M. van den Kerkhof**[1,4], **Jessica Dorning**[5], **Tim G. Rijkhold Meesters**[1], **Patricia van der Woude**[1], **Judith A. Burger**[1], **Tom Bijl**[1], **Riham Ghalaiyini**[1], **Alba Torrents de la Peña**[2], **Hannah L. Turner**[2], **Celia C. Labranche**[3], **Robyn L. Stanfield**[2], **Devin Sok**[6,7,8,9], **Hanneke Schuitemaker**[4], **David C. Montefiori**[3], **Dennis R. Burton**[6,7,8,10], **Gabriel Ozorowski**[2,7,8], **Michael S. Seaman**[5], **Ian A. Wilson**[2,7,8,11], **Rogier W. Sanders**[1,12], **Andrew B. Ward**[2,7,8], **Marit J. van Gils**[1]*

1 Department of Medical Microbiology, Amsterdam Infection & Immunity Institute, Amsterdam UMC, Location AMC, University of Amsterdam, Amsterdam, The Netherlands, 2 Department of Integrative Structural and Computational Biology, The Scripps Research Institute, La Jolla, California, United States of America, 3 Department of Surgery, Duke University Medical Center, Durham, North Carolina, United States of America, 4 Department of Experimental Immunology, Amsterdam Infection & Immunity Institute, Amsterdam UMC, Location AMC, University of Amsterdam, Amsterdam, The Netherlands, 5 Center for Virology and Vaccine Research, Beth Israel Deaconess Medical Center, Harvard Medical School, Boston, Massachusetts, United States of America, 6 Department of Immunology and Microbiology, The Scripps Research Institute, La Jolla, California, United States of America, 7 International AIDS Vaccine Initiative Neutralizing Antibody Center, The Scripps Research Institute, La Jolla, California, United States of America, 8 Consortium for HIV/AIDS Vaccine Development, The Scripps Research Institute, La Jolla, California, United States of America, 9 International AIDS Vaccine Initiative, New York, New York, United States of America, 10 Ragon Institute of MGH, MIT and Harvard, Cambridge, Massachusetts, United States of America, 11 The Skaggs Institute for Chemical Biology, The Scripps Research Institute, La Jolla, California, United States of America, 12 Department of Microbiology and Immunology, Weill Medical College of Cornell University, New York, New York, United States of America

* M.J.vangils@amsterdamumc.nl

**Data Availability Statement:** Atomic coordinates and structure factors of the reported crystal structure have been deposited in the Protein Data Bank (PDB: 7U5G). Cryo-EM reconstructions have

## Abstract

Broadly neutralizing antibodies (bNAbs) have remarkable breadth and potency against most HIV-1 subtypes and are able to prevent HIV-1 infection in animal models. However, bNAbs are extremely difficult to induce by vaccination. Defining the developmental pathways towards neutralization breadth can assist in the design of strategies to elicit protective bNAb responses by vaccination. Here, HIV-1 envelope glycoproteins (Env)-specific IgG⁺ B cells were isolated at various time points post infection from an HIV-1 infected elite neutralizer to obtain monoclonal antibodies (mAbs). Multiple antibody lineages were isolated targeting distinct epitopes on Env, including the gp120-gp41 interface, CD4-binding site, silent face and V3 region. The mAbs each neutralized a diverse set of HIV-1 strains from different clades indicating that the patient's remarkable serum breadth and potency might have been the result of a polyclonal mixture rather than a single bNAb lineage. High-resolution cryo-electron microscopy structures of the neutralizing mAbs (NAbs) in complex with an Env trimer generated from the same individual revealed that the NAbs used multiple strategies to neutralize the virus; blocking the receptor binding site, binding to HIV-1 Env N-linked glycans, and disassembly of the trimer. These results show that diverse NAbs can complement

been deposited in the Electron Microscopy Data Bank and in the Protein Data Bank (PDB: 7ZLK and EMD-14783). The negative stain 3D EM reconstructions are deposited to the Electron Microscopy Data Bank (EMD-14784, EMD-14785, EMD-14786, EMD-14787, EMD-14788 and EMD-14789). The accession numbers for ACS101-103, ACS110-117, ACS120-126 and ACS130-131 BCR sequences are DDBJ/ENA/GenBank: ON098172-ON098203.

**Funding:** This work was supported by the HIV Vaccine Research and Design (HIVRAD) program (P01 AI110657 to A.B.W., I.A.W., R.W.S), the Bill and Melinda Gates Foundation CAVD (OPP1132237 to R.W.S., INV002022 to R.W.S. and OPP1146996 to M.S.S.), OPP002916 to A.B. W., NIH/NIAID (HHSN272201800004C to D.C.M), and the European Union's Horizon 2020 research and innovation program under grant agreement no. 681137 (R.W.S.). R.W.S. is a recipient of a Vici fellowship from the Netherlands Organization for Scientific Research (NWO). J.v.S. is a recipient of a 2017 AMC Ph.D. Scholarship. This research used resources of the SSRL, SLAC National Accelerator Laboratory, which is supported by the U.S. Department of Energy, Office of Science, Office of Basic Energy Sciences under Contract No. DE-AC02–76SF00515. The SSRL Structural Molecular Biology Program is supported by the DOE Office of Biological and Environmental Research, and by the National Institutes of Health, National Institute of General Medical Sciences (including P41GM103393). The funders had no role in study design, data collection and analysis, decision to publish, or preparation of the manuscript.

**Competing interests:** The authors have declared that no competing interests exist.

each other to achieve a broad and potent neutralizing serum response in HIV-1 infected individuals. Hence, the induction of combinations of moderately broad NAbs might be a viable vaccine strategy to protect against a wide range of circulating HIV-1 viruses.

## Author summary

Current behavioral strategies and (non-vaccine) biomedical interventions have been unable to stop the HIV-1/AIDS pandemic. A vaccine will be essential to eradicate the virus from the human population. The focus of HIV-1 vaccine design is to elicit broadly neutralizing antibodies that can block infection of viruses belonging to genetically diverse circulating HIV-1 clades. These bNAbs bind to the HIV-1 envelope glycoprotein (Env) and develop in a subset of HIV-1 infected individuals after several years of infection. Defining the developmental pathways and molecular determinants in *env* sequences that initiated bNAb responses in HIV-1 infected individuals can assist in the design of vaccines to elicit similar type of responses. Here, we isolated various antibody lineages from an HIV-1 infected individual and show that these antibodies target distinct epitopes on HIV-1 Env. None of the antibodies were extremely broad but we found that the antibody lineages each neutralized a different set of HIV-1 viruses from different clades. This suggests that the patient's remarkable serum breadth and potency may have been the result of a complementary polyclonal response. Vaccination strategies to induce combinations of moderately broad antibodies might be a viable approach to protect against infection.

## Introduction

Human immunodeficiency virus 1 (HIV-1) infection is characterized by the high mutability and rapid diversification of the virus driven by a broad set of responses of the host's immune system, i.e. B and T cell responses [1–3]. Infection with one or a few transmitter/founder (TF) virus(es) leads to the development of antibodies able to bind the viral envelope glycoprotein (Env) and neutralize the TF virus or other early viral viruses [2,4], thereby favoring the survival of escape variants. Affinity-matured neutralizing antibodies (NAbs) then pursue the viral escape variants resulting in a complex interplay between viral evolution and antibody maturation. This co-evolutionary process of antibody maturation and virus diversification leads to the development of cross-reactive neutralizing serum responses in 20–50% of HIV-1 infected individuals after several years of infection [5–11]. The isolation and characterization of broadly neutralizing antibodies (bNAbs) from such HIV-1 infected individuals have identified six major Env regions targeted by bNAbs, including the membrane proximal external region (MPER), gp120-gp41 interface, fusion peptide (FP), CD4-binding site (CD4bs), V1/V2-glycan site at the trimer apex, and finally the V3-glycan site [12,13]. Furthermore, two bNAbs have been recently identified targeting a novel bNAb epitope, termed the "silent face" of HIV-1 Env [14,15]. Many bNAbs exhibit unusual genetic and structural features that are not easily accomplished by vaccination; high levels of somatic hypermutation (SHM), long heavy chain complementarity determining region 3 (CDRH3), insertion-deletions events and/or the requirement of a restricted set of VH or VL/VK gene segments. Nevertheless, passively transferred bNAbs have been shown to protect non-human primates from infection [16–18], providing the rationale for attempts to induce bNAbs by vaccination. The genetic and structural

characterization of bNAbs have fueled such efforts, although no vaccine candidate has been able to consistently elicit bNAb responses yet [19].

High anti-Env IgG titers and overall plasma IgG levels correlate with neutralization breadth suggesting that neutralization breadth relies on an extensive and diverse antibody response against Env [20]. In addition, a polyclonal antibody response targeting multiple epitopes has been shown to achieve neutralization breadth in HIV-1 infected adults [9,20–23], but also children [24]. The recently published results of the Antibody Mediated Prevention (AMP) trials imply that an HIV-1 vaccine may likely need to induce a number of (b)NAb specificities capable of neutralizing different sets of viruses, or targeting different subsites within Env, to achieve clinical efficacy [25]. This could be achieved by inducing a polyclonal response against a single epitope, for example the CD4bs, consisting of a number of related (b)NAbs with varying specificities and neutralization profiles. One might also envision a vaccination approach that induces various (b)NAbs lineages against multiple sites on HIV-1 Env, as this could either lead to coverage against a larger portion of circulating HIV-1 strains. Furthermore, a combination of antibodies to different sites could reduce the threshold for neutralization compared to a single site, especially if the antibody potency (and duration) was not sufficient to a single site.

We have previously described a novel CD4bs-targeting NAb ACS101 isolated from an HIV-1 infected elite neutralizer who was enrolled in the Amsterdam Cohort Studies on HIV-1/AIDS [26]. Here, we describe three additional NAb lineages isolated from the same individual, which target the silent face, gp120-gp41 interface and a V3-directed epitope on HIV-1 Env. We show that these four distinct NAb lineages neutralize complementary sets of HIV-1 viruses from different clades and, when combined, achieve considerable neutralization breadth. These findings have implications for HIV-1 vaccine design strategies to induce multiple moderately broad NAbs against multiple sites to protect against HIV-1 infection.

## Results

### Diverse Env-specific B cells were isolated from an HIV-1 infected elite neutralizer

Individual H18877 was infected with a clade B HIV-1 variant and categorized as an elite neutralizer because of the extreme potent and broad neutralizing serum response that was elicited early in infection (Fig 1A) [5,27]. Longitudinal *env* sequences from months 2, 8, 10, 13, 18, 27, 36 and 49 post-SC were previously isolated from this patient, of which five early *env* sequences were used to generate the AMC009 SOSIP trimer protein [5,28]. We sought to identify the (b) NAb specificities in this individual and sorted Env-specific B cells from peripheral blood mononuclear cells (PBMCs) at months 2, 11, 24, 36 and 49 post-SC (Fig 1A–1B). B cells from all timepoints were sorted with fluorescently labeled clade A BG505 SOSIP and clade A 94UG103 gp120 proteins and either included the autologous clade B AMC009 SOSIP (months 2, 11, 24 and 49) or clade C MGRM C026 gp120 (month 36) protein (Fig 1B) [26]. In total, we sorted 751 Env-specific memory B cells from this HIV-1 infected elite neutralizer.

We obtained 314 heavy chain (HC) B cell receptor sequences (BCR) from the sorted HIV-1 Env-specific B cells. The sorted B cells used a wide variety of VH genes for their BCR, demonstrating the heterogeneity of the antibody response in this individual (Fig 1C). The corresponding light chain (LC) sequences were polymerase chain reaction (PCR) amplified and productive HC and LC pairs were cloned into expression vectors to generate monoclonal antibodies (mAbs). When successfully expressed, mAbs were tested in an enzyme-linked immunosorbent assay (ELISA) for binding to AMC009 SOSIP and subsequently screened for neutralization against the pseudotyped AMC009 virus, which was generated from one of the five viral clones isolated at two months post-SC [28], and a small panel of heterologous viruses.

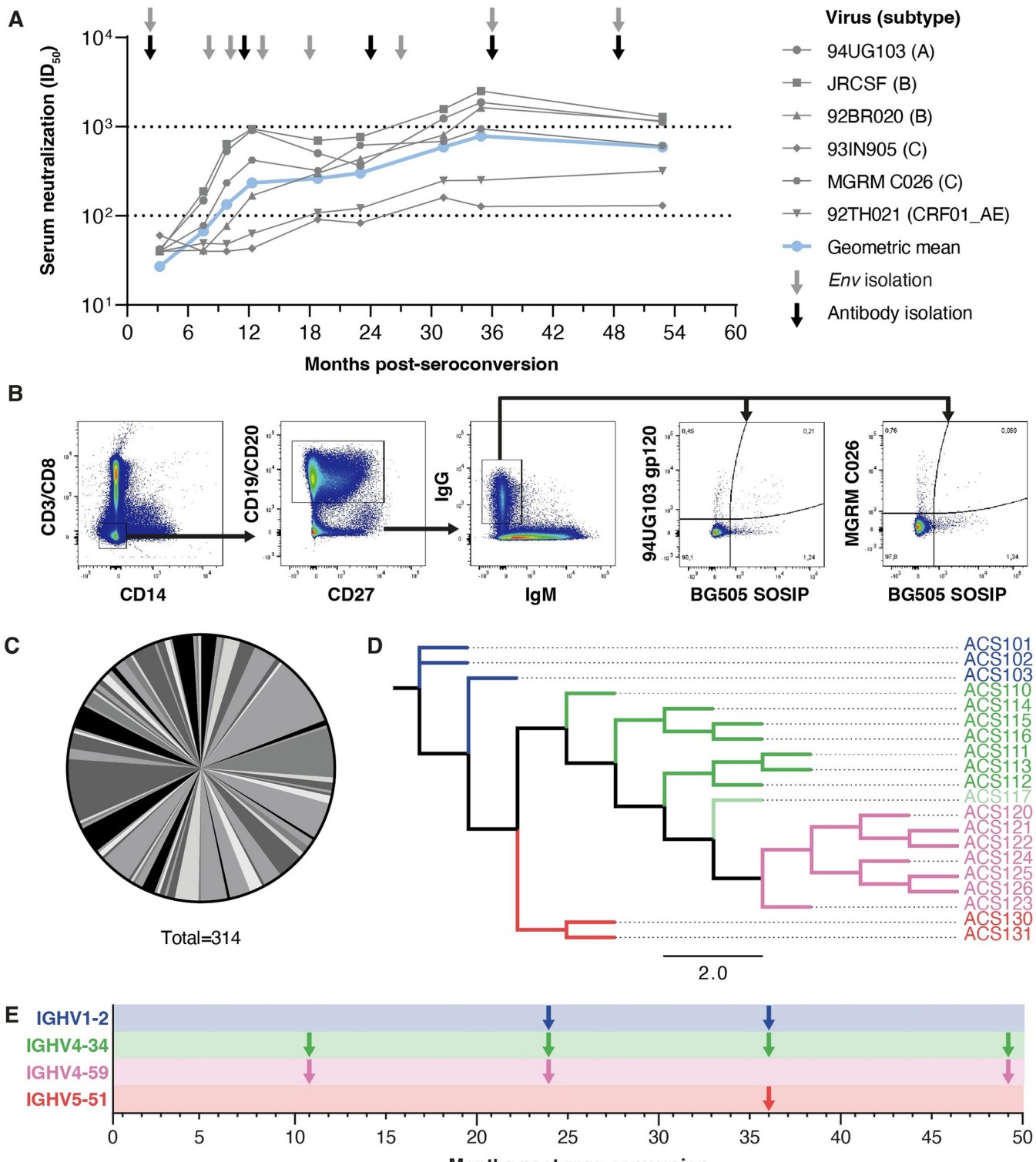

**Fig 1. Isolation of four distinct antibody lineages from an HIV-1 infected elite neutralizer using Env-specific B cell sorting. (a)** Longitudinal serum neutralization of viruses from different clades by individual 18877. The geometric mean is given in blue and arrows indicate timepoints where virus (grey) and PBMCs (black) were isolated to obtain *env* and BCR sequences, respectively. (**b**) B cells were selected by fluorescence-activated cell sorting (FACS) using fluorescently labeled Env proteins. The gating strategy for the isolation of B cells from PBMCs taken at month 36 post-SC is shown here as an example. (**c**) Pie-chart displaying the VH gene usage of the isolated BCR sequences (n = 314). Each grey-color coded slice represents a VH gene and the size of the slice is proportional with the amount of BCRs derived from this gene. (**d**) Phylogenetic analysis of the HC VDJ sequences of four isolated antibody lineages to compare

the evolutionary relationship between the various BCRs. The names of the corresponding mAbs are listed on the right. Colors indicate to what lineage the mAbs belong to: IGHV1-2 (blue), IGHV4-34 (green), IGHV4-59 (pink) and IGHV5-51 (red). ACS117 is colored slightly lighter compared to the other members of the IGHV4-34 lineage because ACS117 most likely originated from a different B cell precursor. Scale depicts the nucleotide substitutions per site. (**e**) Timeline with arrows indicating the timepoints that members from the four different antibody lineages were obtained.

Based on these results, we selected three antibody lineages derived from the IGHV4-34, IGHV4-59 and IGHV5-51 germline genes for further characterization, as these were the antibodies that showed neutralization capacity (S1–S3 Figs). For all subsequent analyses, we also included the information from the IGHV1-2-derived IOMA-class CD4bs-targeting mAbs that were previously isolated at months 24-, and 36 post-SC from the same individual [26]. Phylogenetic analysis of the HC variable regions showed that, within the antibody lineages, genetic distances were substantial, most likely a result of the extensive SHM acquired during multiple years of infection (Figs 1D and S1). Members from the four antibody lineages were isolated from at least two different time points post-SC, except for the two mAbs of the IGHV5-51 lineage that were isolated solely at 36 months post-SC (Figs 1E and S1).

## Distinct antibodies achieve neutralization breadth by complementarity

We previously isolated three novel CD4bs-targeting NAbs (ACS101-ACS103) with moderate neutralization breadth from this individual that were derived from the IGHV1-2*02 germline, and included a normal-length (eight residue) LC complementarity determining region 3 (CDRL3) and fewer SHM compared to VRC01-class bNAbs (Figs 2A, S1 and S2A) [26]. While these NAbs could be classified as novel members of the IOMA-class of CD4bs bNAbs, it remained uncertain given their diversity whether these NAbs were derived from the same precursor, or represented three unrelated IOMA-class NAbs within the same patient [26].

The second lineage consisted of seven members (ACS110-ACS116) with HCs and LCs derived from the IGHV4-34 and IGKV2-24 gene segments with CDRH3 and CDRL3 lengths of 14 and 9 amino acids, respectively (S1 and S2B Figs). In addition, we isolated an eighth member (ACS117) that utilized similar IGHV and IGKV gene segments, but included a CDRH3 of 13 amino acids and did not share many SHM with the other mAbs, and thus possibly originated from a different B cell precursor (S1 and S2B Figs). Members were isolated from four different timepoints with varying SHM levels ranging from 14–26 amino acids in the IGHV gene segments and 4–11 amino acids in the IGKV gene segments. Generally, SHM levels were higher in the mAbs isolated at later timepoints. When tested against a panel of viruses representing HIV-1 global diversity, four mAbs (ACS113, ACS114, ACS115 and ACS116) neutralized the clade B TRO.11 virus, whereas mAb ACS117 neutralized the clade AC 246-F3 virus (S3A–S3B Fig). In addition, various members neutralized the autologous AMC009 virus. Negative-stain, single particle electron microscopy (NS-EM) of ACS110, ACS114 and ACS117 Fabs in complex with AMC009 SOSIP revealed that they targeted the silent face of HIV-1 Env (Fig 2A).

A third clonal family (ACS120-ACS126) was derived from PBMCs isolated at months 11-, 24-, and 49-months post-SC and consisted of seven members with HCs and LCs utilizing the IGHV4-59 and IGLV3-21 gene segments (S1 and S2C Figs). The mAbs exhibited a relatively low level of SHM ranging from 10–17 and 3–8 amino acids in their IGHV and IGLV gene segments with CDRH3 and CDRL3 lengths of 18 and 11 amino acids, respectively. Furthermore, mAbs isolated at later timepoints contained higher SHM levels than those isolated at earlier timepoints. Various members of this lineage neutralized the clade C Ce1176 and autologous AMC009 viruses, albeit weakly (S3A Fig). NS-EM of three Fabs (ACS122, ACS124 and ACS125; one from every timepoint) in complex with the autologous AMC009 SOSIP trimer showed that they bound the gp120-gp41 interface (Fig 2A).

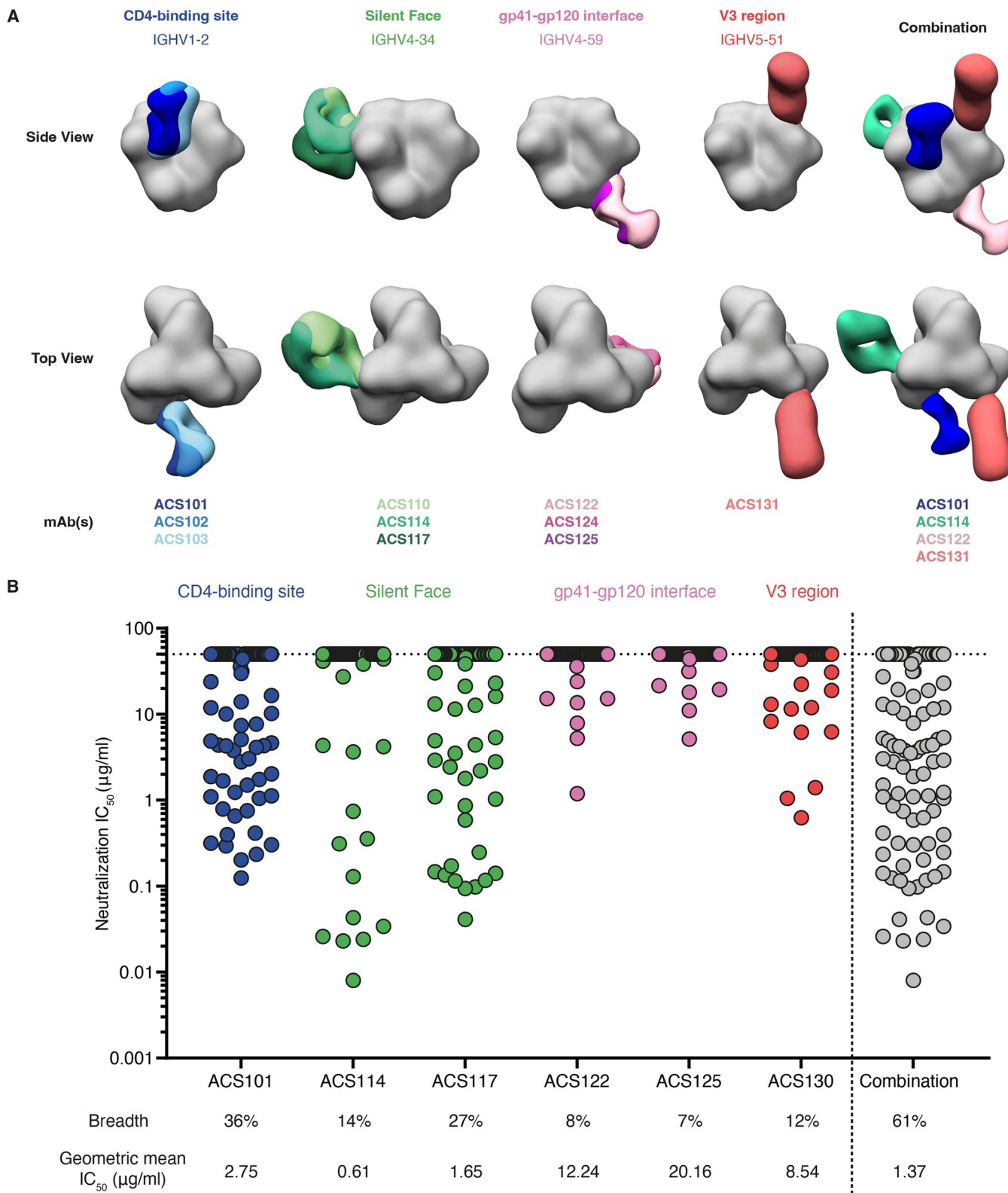

**Fig 2. Complementary neutralizing antibody lineages target four distinct epitopes on HIV-1 Env. (a)** Epitope mapping by negative-stain electron microscopy. Fabs or IgG were complexed with the AMC009 SOSIPv5.2 trimer. Composite figures showing representative mAbs of the four antibody lineages targeting the CD4-binding site (blue), silent face (green), gp120-gp41 interface (pink) and V3 region (red). The mAbs included in the NS-EM analysis are shown at the bottom and colors correspond to those shown in the composite figure. Fabs were segmented, colored and are shown relative to a reference trimer

(grey) for visualization and comparison purposes. (**b**) Neutralization potency of representative mAb(s) from every lineage against a multiclade 119-virus panel. The mAbs were tested at a starting concentration of 50 μg/ml. Neutralization $IC_{50}$ is plotted as the antibody concentration (μg/ml) that inhibits 50% of viral infectivity. Each symbol represents a unique HIV-1 isolate, with the different colors indicating the epitope targeted by the mAbs. Breadth is indicated below as the percentage of viruses that were neutralized with the geometric mean in μg/ml below. We also provide an estimated neutralization breadth of all mAbs combined.

Lastly, two mAbs were isolated utilizing the IGHV5-51*01 and IGLV1-51*01 gene segments with CDRH3 and CDRL3 lengths of 12 and 10 amino acids, respectively (S1 and S2D Figs). ACS130 and ACS131 contained 11 and 12 amino acid substitutions in their HC whereas SHM levels in the LC was 12 and 7 amino acids, respectively. Furthermore, ACS130 weakly neutralized the clade G virus X1632 (S3A–S3B Fig). NS-EM initially specified the epitope of these mAbs to a region surrounding the V1/V2/V3 region of HIV-1 Env (Fig 2A) and were later shown to target a V3-directed epitope. Interestingly, the IGHV5-51 gene utilized by these mAbs is commonly used by mAbs targeting the V3 [29,30].

The various antibody lineages each neutralized different viruses and when combined could neutralize seven out of nine viruses from the global panel, suggesting a complementary effect in neutralization breadth (S3A–S3B Fig). To study this further, we tested a selection of NAbs from every lineage against a multiclade panel (n = 119) of HIV-1 viruses to determine their complementary neutralization breadth (Figs 2B and S4). The CD4bs NAb ACS101 neutralized 36% of tested viruses with a geometric mean of the half-maximal inhibitory concentration ($IC_{50}$) of 2.8 μg/ml and exhibited greater breadth against clades A and B compared to other clades. The silent face NAbs ACS114 and ACS117 had a neutralization breadth of 14% and 27% with geometric mean $IC_{50}$'s of 0.61 and 1.7 μg/ml, respectively. Interestingly, NAb ACS117 was better at neutralizing viruses from clade CRF01_AE compared to other NAbs. In addition, the gp120-gp41 interface NAbs ACS122 and ACS125 were less broad and potent and neutralized 8% and 7% of tested viruses with a $IC_{50}$ geometric mean of 12 and 20 μg/ml, respectively. The V3-targeting NAb ACS130 neutralized 12% of tested viruses with a geometric mean $IC_{50}$ of 8.5 μg/ml. Although none of the NAbs were extremely potent or broad, when combined, they neutralized 61% of all tested viruses with a geometric mean $IC_{50}$ of 1.4 μg/ml.

Biolayer interferometry demonstrated that antibodies from every lineage could bind AMC009 SOSIP trimer simultaneously (S5A Fig). To determine if a polyclonal mixture of NAbs could increase neutralization potency, we tested a combination of NAbs, consisting of one member from every lineage, against a multiclade set of 15 HIV-1 Env pseudoviruses. This panel was based on positive hits from the large (n = 119) multiclade virus panel. Two antibody mixtures that included NAbs ACS101, ACS122, ACS130 and ACS114 or ACS117 were tested. The antibody mixtures neutralized the set of viruses with similar $IC_{50}$'s compared to the $IC_{50}$'s of the most potent antibodies in the mixture when tested individually against the same set of viruses. Thus, the antibody mixtures did not increase potency of the tested viruses and the neutralizing effect appeared mostly to be additive, with no significant evidence of synergistic enhancement of neutralization (S5B Fig). These findings reveal that, during the course of HIV-1 infection in this patient, several antibody lineages developed that were directed to different antigenic regions on HIV-1 Env and their combination was able to achieve neutralization breadth.

## Silent face antibodies primarily interact with N-linked glycans

The characterization of NAb epitopes and their evolution over time using longitudinal *env* sequences should help to understand the virus-host interactions within this individual, and could inform HIV-1 vaccine design to induce similar NAb responses by vaccination. We have

previously elucidated the structural basis for ACS101's neutralization breadth and the evolution of its CD4bs epitope over time [26]. To better understand the epitope-paratope interaction of the additional antibody lineages isolated from this patient, we also performed single-particle cryo-electron microscopy (cryo-EM) imaging of AMC009 SOSIP in complex with the gp120-gp41 targeting ACS122 and silent face-targeting ACS114 Fabs. The cryo-EM complex of ACS122 and ACS114 Fabs bound to AMC009 SOSIP was reconstructed to a resolution of ~4.0 Å and contained three ACS114 Fabs and two ACS122 Fabs interacting with the AMC009 SOSIP trimer (S6A–S6C Fig). We also solved a 1.7 Å-resolution crystal structure of the ACS122 Fab but was unable to crystallize ACS114 Fab by itself (S6D Fig).

The ACS122-ACS114-AMC009 cryo-EM reconstruction demonstrated that the ACS114 Fab interacts predominantly with the N295, N301, N332 and N262 glycans, and recognition is primarily mediated via the HC with only minor contacts of the LC (Fig 3A–3B). Specifically, ~92% (1459 Å$^2$) of the total average buried surface area (BSA) is contributed by ACS114 HC, 82% (1190 Å$^2$) of which consists of interactions with Env glycans. The majority of the interactions are with the N295 glycan and contact is mediated via residues in both the ACS114 HC and LC (Fig 3B). Interestingly, when tested against a panel of JRCSF pseudovirus mutants, we found that ACS114 critically depends on the N295 glycan for neutralization (S7A Fig). While ACS114 interacts primarily with N-linked glycans, the antibody also makes contact with residues in the gp120 C4 region (Fig 3C). The epitope of ACS114 overlaps to a large extent with the epitopes of silent face-targeting bNAbs VRC-PG05 and SF12; however, the epitope of ACS114 is located slightly higher up on the trimer showing subtle differences in recognition of the silent face epitope (Fig 3D). In contrast, the eighth member of this lineage (ACS117) approached the silent face of HIV-1 Env more akin to bNAbs VRC-PG05 and SF12, and also exhibited similar potency and neutralization breadth as VRC-PG05 [14] (Figs 2B and S7B-S7C). In summary, we isolated additional members that can be added to the rare class of bNAbs that target the silent face and show that these NAbs predominantly interact with N-linked glycans on HIV-1 Env.

## gp120-gp41 interface antibodies destabilize HIV-1 Env

The high-resolution cryo-EM structure revealed that ACS122 binds an epitope surrounding the gp120-gp41 interface with an average buried surface area (BSA) of 1246 Å$^2$, 86% (1068 Å$^2$) of which is contributed by the ACS122 HC (Figs 4A and S8A). The majority of the interactions are via ACS122 HC variable loops that interact with the gp41 fusion peptide proximal region (FPPR) and HR2 region (Figs 4A and S8A). Most of the residues that ACS122 contacts in Env are highly conserved, e.g., S528 (99.53%) G531 (99.83%) A532 (99.49%) T536 (90.45%) N625 (97.04%), including the N-linked glycan at position N88 (99.53%) (S8A Fig). In contrast, ACS122 CDRH3 makes extensive contact with N624 in gp41, which is only present in 31.5% of HIV-1 strains. Interestingly, NS-EM and cryo-EM 2D-class averages of ACS122 in complex with AMC009 SOSIP trimer showed that ACS122 induced dissociation of AMC009 trimer into individual gp120-gp41 protomers (Fig 4B). Dissociation of the AMC009 SOSIP trimer did not occur spontaneously when the trimer was left overnight at room temperature in the absence of ACS122. This observation suggests that the Fab was responsible for the observed trimer dissociation. Moreover, the C-terminal part of gp41 was disordered in the cryo-EM map for two gp41 subunits suggesting that CDRH3 and CDRL3 of ACS122 cause the HR2 region of the adjacent gp41 to relocate (Fig 4A and 4C). For a third gp41 monomer to be bound, it likely requires more opening at the base of the trimer leading to dissociation and trimer degradation. The cryo-EM structure also demonstrated that ACS122 uses its long CDRH3 to disrupt the tryptophan clasp at the base of the trimer, akin to bNAb 3BC315 [31] and

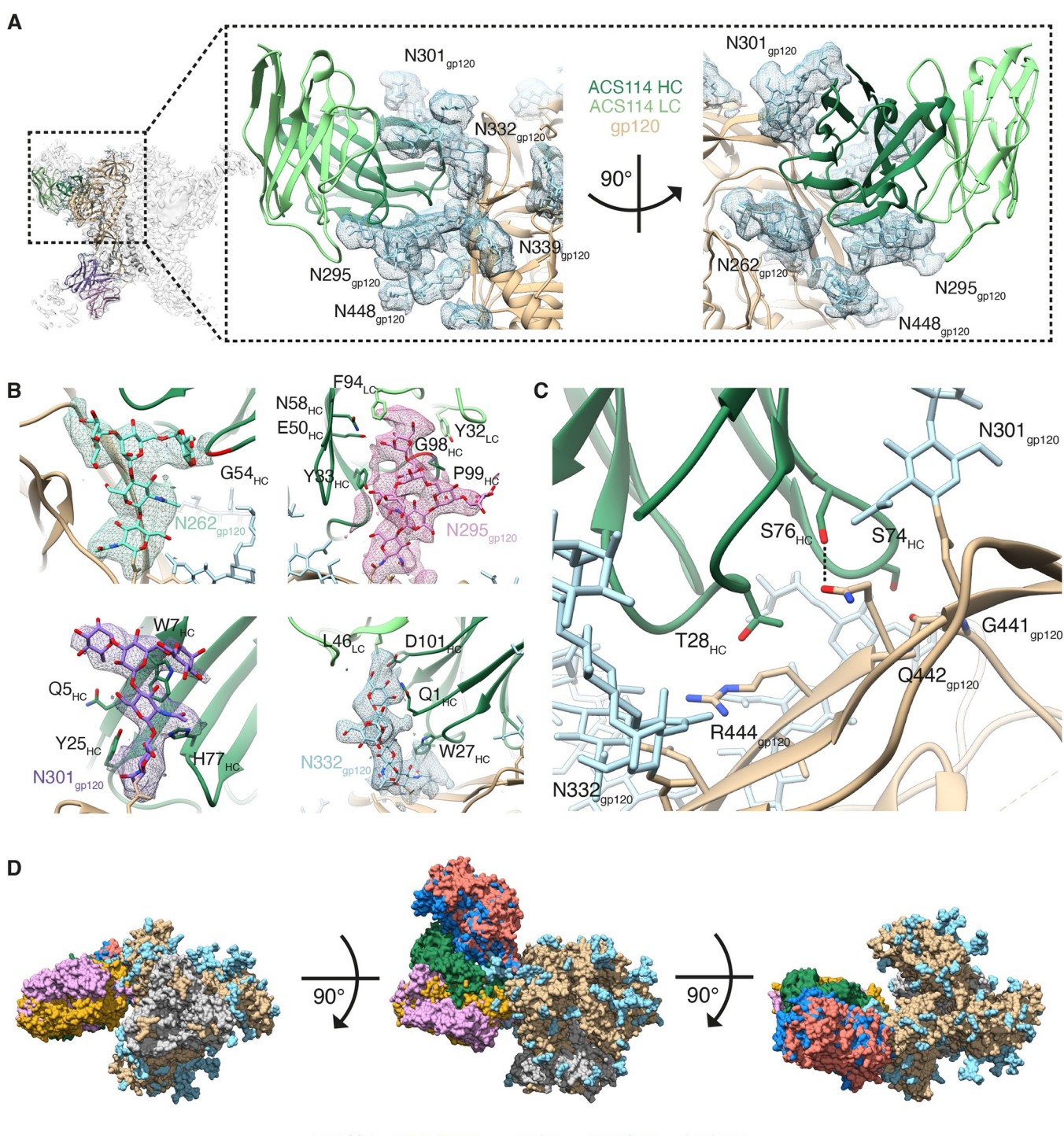

**Fig 3. ACS114 targets the silent face and primarily interacts with *N*-linked glycans on HIV-1 Env.** (**a**) Magnified view of the ACS114 epitope shown as ribbons. ACS114 HC and LC are color-coded as indicated. Glycans are shown as sticks and contoured by the density of the cryo-EM map at 4σ. Gp120 is color-coded beige. (**b**) Interaction of ACS114 with the N262, N295, N301 and N332 glycans. Glycans are shown as sticks and contoured by the density of the cryo-EM map at 4σ. Amino acids that interact with the respective glycans are highlighted. (**c**) Interaction of ACS114 HC with gp120 C4 region. Amino acid interactions between ACS114's HC and gp120's C4 are highlighted and based on the density in the cryo-EM map. Predicted hydrogen bonds are shown with a distance <3.2 Å. (**d**) Comparison of ACS114 to V3-targeting bNAb PGT124 (PDB:6MCO) and 10–1074 (PDB:5T3Z) and silent face-targeting bNAbs VRC-PG05 (PDB:6BF4) and SF12 (PDB:6OKQ). Fabs and trimer are shown as a surface representation. Fabs are colored coded as indicated. The gp41 and gp120 subunits are depicted in grey and light brown, respectively, with the N-linked glycans in light blue. The structures were aligned relative to gp120.

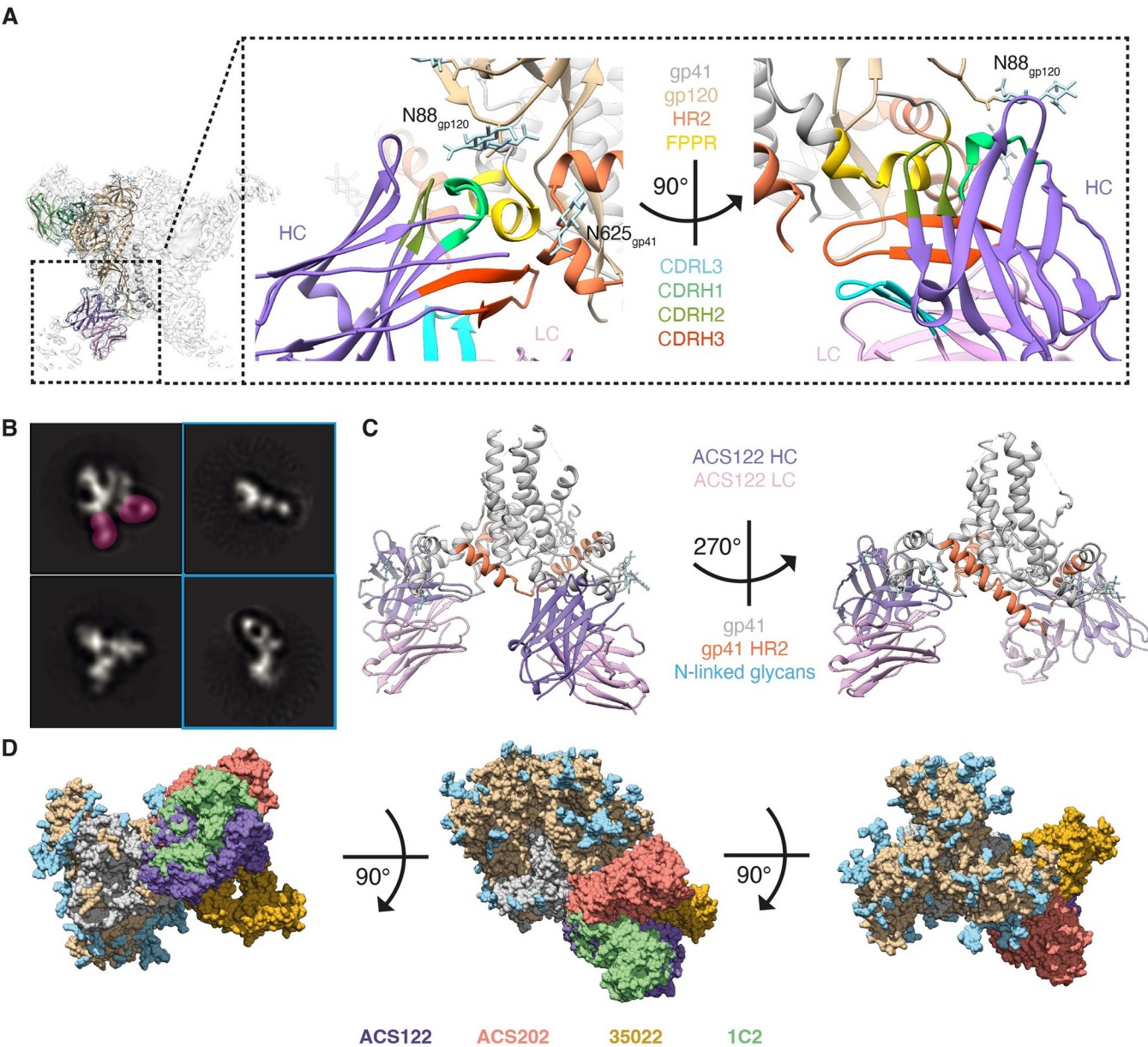

**Fig 4. ACS122 destabilizes HIV-1 Env by binding the gp120-gp41 interface.** (**a**) Magnified view of the ACS122 epitope shown as ribbons. The different regions of ACS122 HC, ACS122 LC and AMC009 SOSIP trimer are color-coded as indicated. Glycans are shown as sticks. (**b**) NS-EM analysis of ACS122 bound to the AMC009 SOSIP trimer. Representative 2D class averages are shown with two ACS122 Fabs bound to the trimer (left upper corner). ACS122 is highlighted in pink. 2D class averages with dissociated trimer bound by ACS122 Fabs are boxed in blue. (**c**) Magnified view of two ACS122 Fabs bound to the gp41 subunits shown as ribbons. The gp120 subunits are hidden for visualization purposes. Part of the HR2 region is colored in orange. (**d**) Comparison of ACS122 versus the FP-targeting bNAb ACS202 (PDB:6NCP) and gp120-gp41 interface targeting bNAbs 35022 (PDB:5W6D) and 1C2 (PDB:6PEH). Fabs and trimer are shown as a surface representation. Fabs are colored coded as indicated. The gp41 and gp120 subunits are depicted in grey and light brown, respectively, with the N-linked glycans in light blue. The structures were aligned relative to gp41.

vaccine-elicited rabbit bNAb 1C2 that both cause the trimer to dissemble [32] (S8A Fig). We then compared the angle of approach of ACS122 with Env to bNAbs targeting similar regions, e.g., ACS202 [33], 35022 [34] and bNAb 1C2 that was isolated from an HIV-1 Env immunized rabbit [32]. We found that ACS122 approaches Env at a different angle compared to ASC202 and 35022, which may explain the lower potency and/or breadth of ACS122

compared to the gp41-gp120 interface and fusion peptide bNAbs 35O22 and ACS122, respectively (Figs 4D and S8B). Moreover, ACS122 contains lower levels of SHM in its HC and LC (24 and 10 nucleotide substitutions respectively) compared to 35022 (96 and 71 nucleotide changes in HC and LC, respectively), and lacks the 8 amino-acid insertion in HC framework 3 that characterizes 35O22 [34,35]. ACS122 contacts Env in a similar way to 1C2 but is far less broad compared to 1C2, which was capable of moderately neutralizing 87% of isolates in a 208-virus panel [32]. Whereas 1C2 stabilizes the N88 glycan with its CDRH1 and HC framework 3 in a position close to N625, ACS122 solely interacts with the N88 glycan with its CDRH1. This suggests that 1C2 may be better at accommodating the N88 glycan compared to ACS122, possibly explaining the difference in neutralization breadth between these two antibodies. These findings support the notion that NAbs targeting the gp120-gp41 interface may destabilize HIV-1 Env as a mechanism to neutralize the virus.

## V3-directed antibodies depend on specific residues in the V3 loop

NS-EM 2D class averages demonstrated that ACS130 and ACS131 bound to AMC009 SOSIP trimer with a stoichiometry of three, two or one Fab(s) to one Env trimer (Fig 5A)., Moreover, from all the particles picked for 2D class averaging, only a subset was bound by Fabs and 3D reconstruction resulted in maps of poor quality. To better understand the exact epitope of these two mAbs, we performed a neutralization assay with a panel of JRCSF pseudoviruses with mutations in the V3 loop of Env (Fig 5B–5C). Neutralization by both mAbs was knocked-out for the JRCSF P313A, R315A and R315Q mutants. In addition, the P310A mutation also knocked-out neutralization for mAb ACS130. The involvement of V3 residues might explain why these mAbs did not bind well to the AMC009 SOSIP.v5.2 trimer as visualized by NS-EM, as the SOSIP.v5.2 design includes a A316W substitution to prevents exposure of the V3 loop [36]. Combined, these data show that ACS130 and ACS131 bind to an epitope involving residues in the V3 loop of HIV-1 Env.

## The CD4-binding site and silent face antibody lineages drove HIV-1 Env evolution

To obtain more insight into the co-evolutionary process of antibody maturation and virus diversification, we studied the viral evolution of longitudinal *env* sequences from this patient and produced several of these Env variants as SOSIP trimers [5,26]. We have previously showed that an N276-lacking virus might have induced the development of the CD4bs response in this individual and that the virus exhibited an early viral escape pathway from antibodies targeting the CD4bs (Figs 6 and S9A) [26].

The high-resolution cryo-EM structure revealed that ACS114 primarily interacts with the N-linked glycans on HIV-1 Env, in particular with the N295 glycan. In one *env* sequence from 8-months post-SC and, in the majority of sequences from later timepoints, the N-linked glycosylation sequon at position 295 was deleted by a variety of substitutions; N295D, N295T, N295Y and T297I (Fig 6). The potential N-linked glycosylation sites (PNGS) for the other glycans contacted by ACS114, i.e. N262, N301 and N332, remained intact over the course of the study, except for the N332 PNGS, which was mutated in sequence 36M.1A8 that contained the S334N substitution (Fig 6). The virus also introduced mutations, starting 8-months post-SC and onwards, in the gp120 C4 region, including at positions 442 and 444 that interact with the HC of ACS114 (Fig 6). Specifically, the R444N mutation that is found starting from 27-months post-SC introduces a N-linked glycosylation sequon at position 444 that would clash with ACS114 HC based on the cryo-EM structure. Surprisingly, ACS114 is still able to bind to SOSIP variant 27m.2C4 that contains the N444 and N448 PNGS (Fig 7). One explanation for

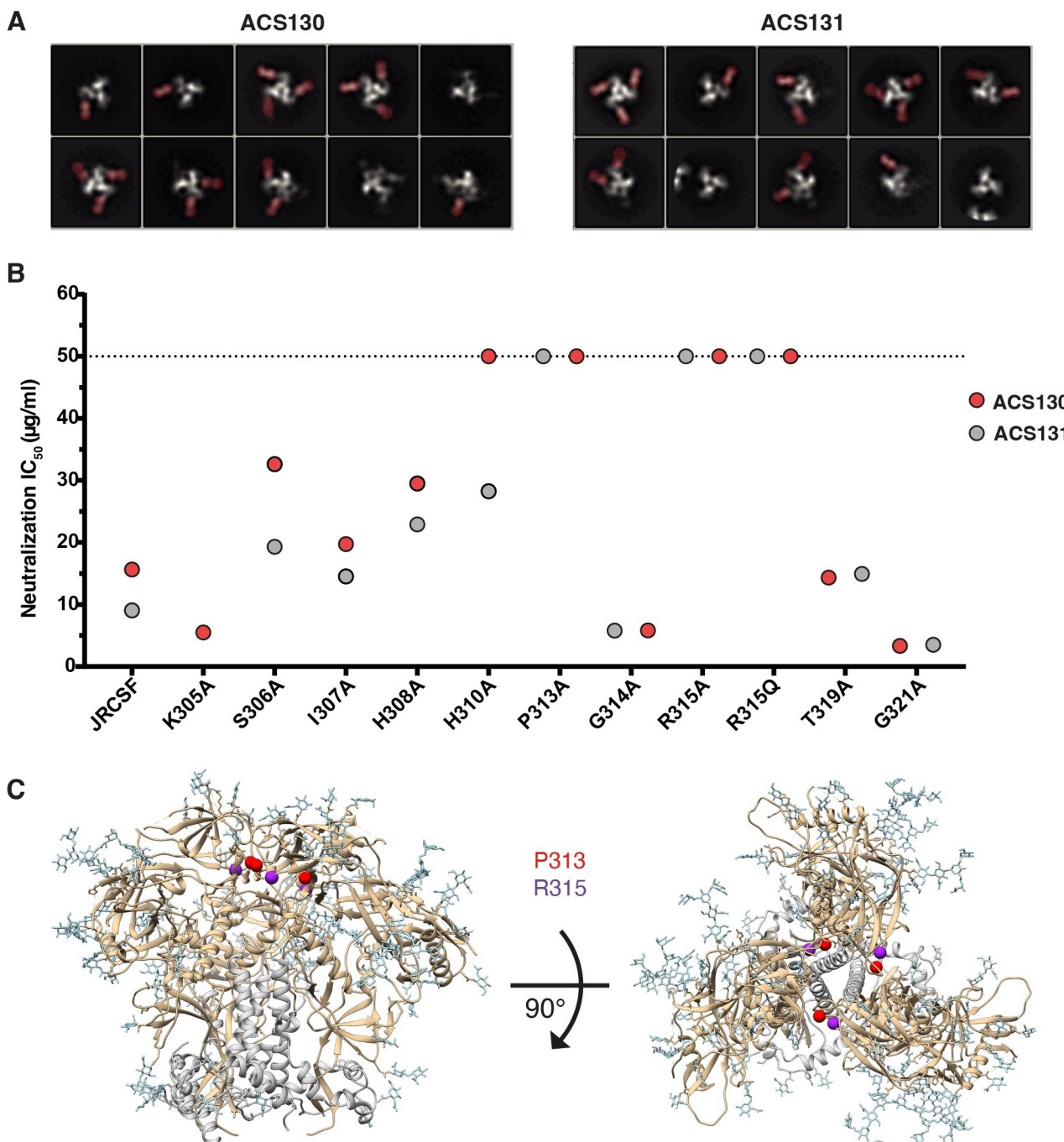

**Fig 5. ACS130 and ACS131 depend on residues in the V3 region of HIV-1 Env for neutralization.** (**a**) Representative NS-EM 2D class-averages of ACS130 and ACS131 bound to AMC009 SOSIP. (**b**) Neutralization potency (antibody concentration (μg/ml) that inhibits 50% of viral infectivity ($IC_{50}$) of ACS130 (red) and ACS131 (grey) against a panel of JRCSF mutants. The different mutants are depicted along the horizontal axis. The mAbs were tested at a starting concentration of 50 μg/ml. (**c**) Ribbon representation of the AMC009 SOSIP trimer with residues P313 (red) and R315 (purple) highlighted. The gp41 and gp120 subunits are depicted in grey and light brown, respectively, with the N-linked glycans in light blue sticks.

the observed binding may be the underoccupancy of the N444 PNGS, as multiple PNGS in close proximity of each other were shown to effect each other's occupancy on SOSIP trimers [37]. Several of the silent face-targeting mAbs from this elite neutralizer bound well to early

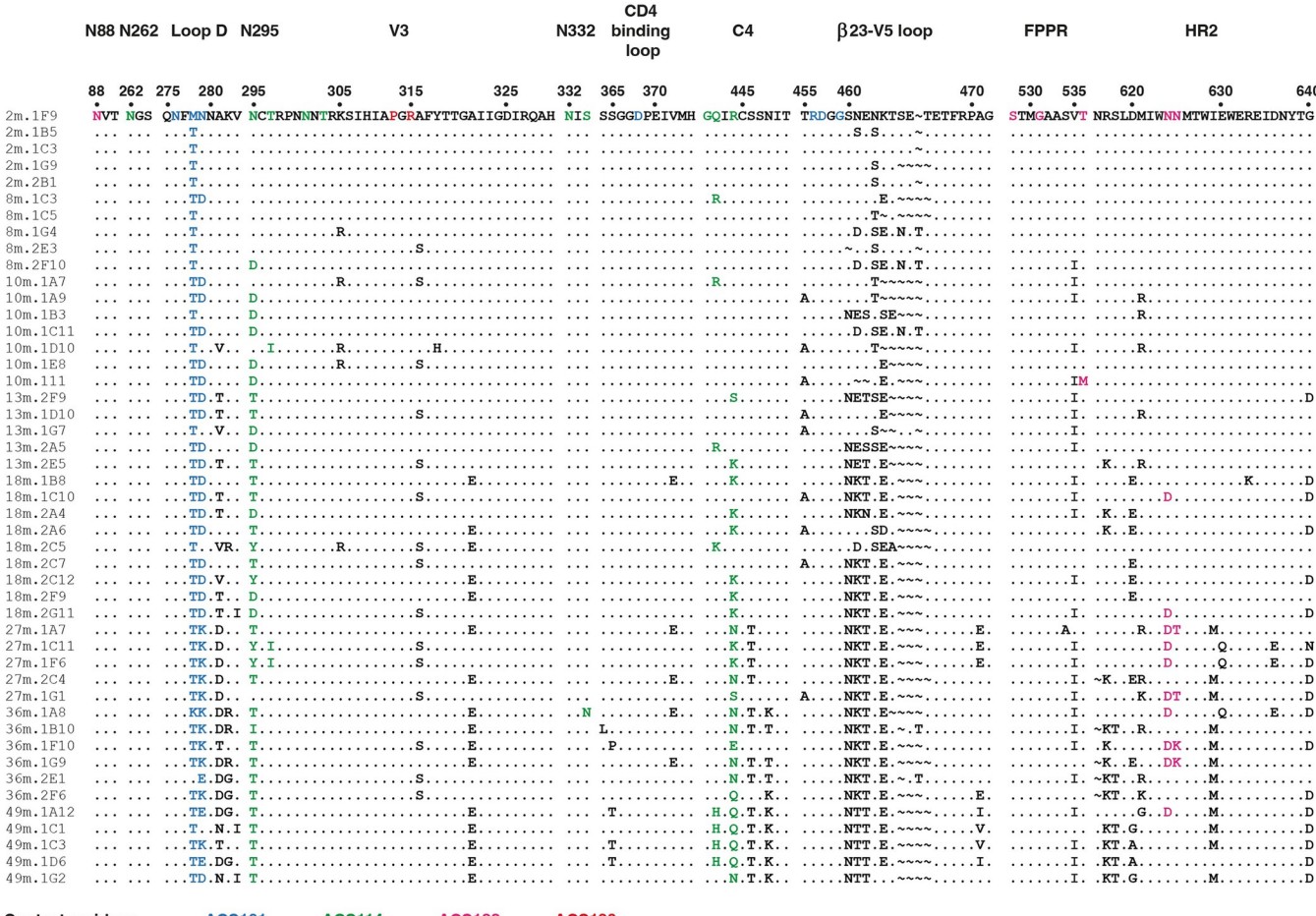

**Fig 6. HIV-1 Env evolution of the CD4-binding site, silent face, gp120-gp41 interface and V3 epitopes targeted by the various NAbs.** Sequence alignment of longitudinal Envs from donor H18877 with residues surrounding the epitopes of all four antibody lineages are shown. Contact-residues of ACS101, ACS114, ACS122 and ACS130 revealed by high-resolution cryo-electron microscopy structures and neutralization assays are highlighted in different colors as shown below. Amino acids are numbered according to HxB2.

Env SOSIP variants including 10M.1A7 that still included the PNGS at position 295, but not to those based on *env* sequences from 10-months post-SC, which lack the N295 glycan (Figs 7 and S9B). Together these data suggest the silent face lineage developed between ~2-, and 8-months post-SC and was followed by viral escape and antibody maturation to accommodate changes in the epitope.

The silent face-targeting NAb ACS117 did not bind well to early Env variants, including those containing the N295 PNGS, suggesting that this member was induced later during infection. Interestingly, ACS117 binding was completely knocked-out for the 36M.1A8 variant that lacked the N448 glycan (Figs 7 and S9B). This finding is consistent with the ACS117 epitope overlapping with those of VRC-PG05 and SF12 and their dependence on the N448 glycan for neutralization [14,15]. The majority of *env* sequences from timepoints 36-, and 49-month post-SC contained a N448K or N448T substitution deleting the PNGS for the N448 glycan, suggesting a viral escape pathway to neutralization by NAb ACS117 between months ~27-, and 36-months post-SC (Fig 6).

Sequence analysis of the longitudinal *env* sequences showed that the virus mutated positions within the FPPR and HR2, two regions that are contacted by the gp120- gp41 interface

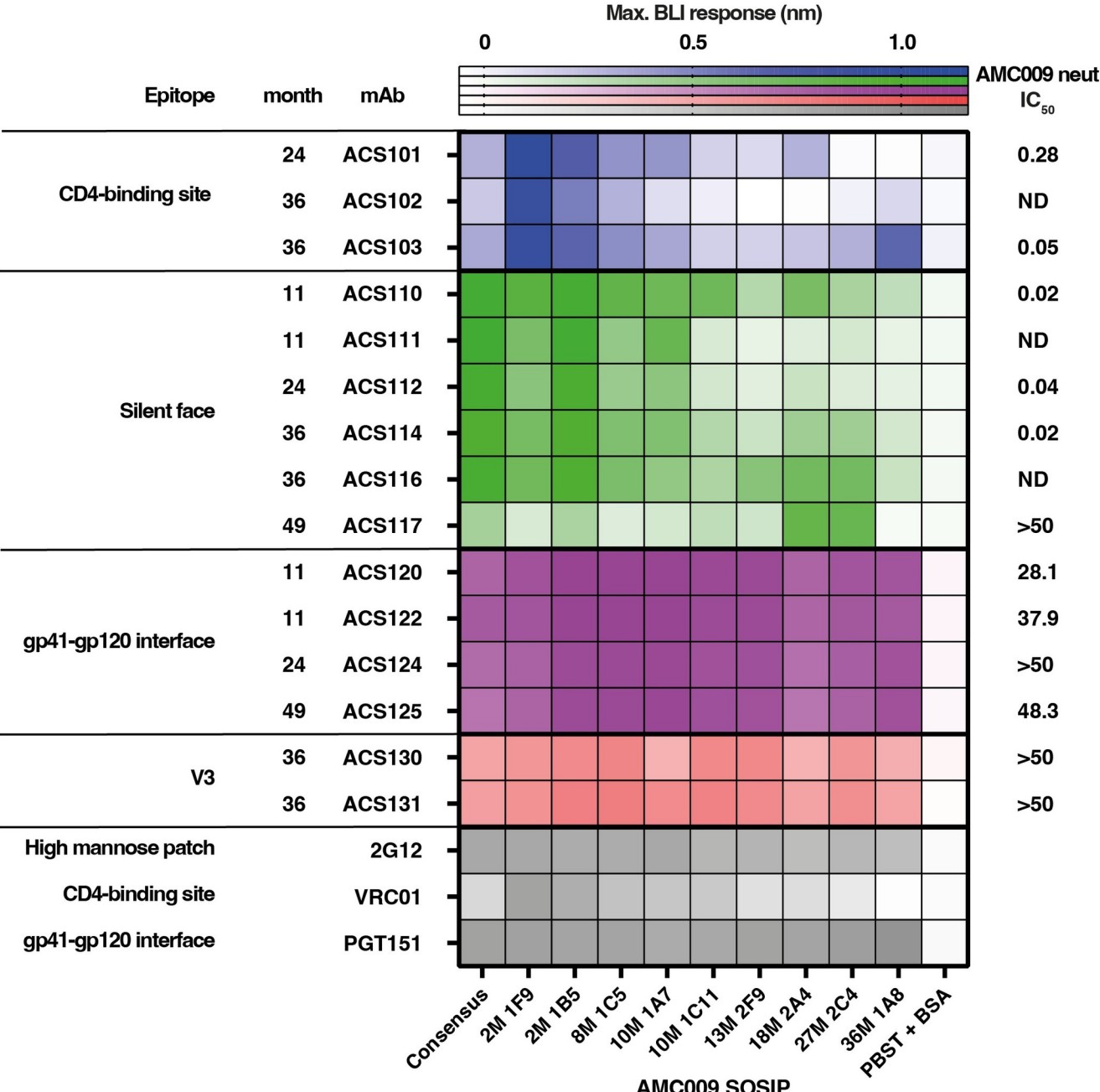

**Fig 7. Binding of mAbs to longitudinal Env using bio-layer interferometry (BLI).** Heat map representing the maximum values of the binding curves measured by BLI. The bNAbs 2G12, VRC01 and PGT151 were used as controls. Neutralization $IC_{50}$ (µg/ml) of the AMC009 virus (2M 1B5) is included on the right side of the figure for comparison purposes.

NAb ACS122, at 8 months post-SC and later timepoints (Fig 6). The mAbs from the gp120-gp41 interface lineage bound equally to all tested Env variants, with only a minor decrease in binding for Env variants from later timepoints compared to those isolated early during infection (Figs 7 and S9C). Based on these results, we were unable to determine the exact time when the gp120-gp41 interface lineage was initiated. However, the viral escape by various substitutions and deletions at positions in Env surrounding the ACS122 epitope may indicate

viral escape between ~2 and 8 months post-SC by additional family members of the gp120-gp41 lineage that were not isolated in this study.

Few mutations were introduced in the V3 region of the longitudinal *env* sequences and V3-targeting mAbs ACS130 and ACS131 bound well to all tested Env variants from different timepoints suggesting that these NAbs did not drive evolution within the V3 (Figs 7 and S9C). Based on these data and the availability of only two members, ACS130 and ACS131, we were unable to determine when this lineage was initiated. Moreover, in general members within all lineages that were isolated at later timepoints bound with higher affinities to the AMC009 SOSIP (S10 Fig and S3 Table), indicating affinity maturation after recognition of the early virus. Taken together, analysis of longitudinal *env* sequences and biolayer interferometry binding experiments suggest that the first wave of NAbs targeted the CD4bs, followed by a second wave of mAbs targeting the silent face and gp120-gp41 interface of HIV-1 Env.

## Discussion

Infection with HIV-1 leads to the development of a wide range of antibodies including those that can neutralize early viral variants [2,4]. The co-evolutionary process of viral escape and antibody maturation over a long period shapes these antibodies into bNAbs in a subset of HIV-1 infected individuals [5–11]. Identifying the molecular determinants in *env* sequences that initiated bNAb responses in these individuals can assist in the design of vaccines to elicit similar type of responses. In this study, we isolated several NAb lineages from an HIV-1 infected elite neutralizer, and found that they target multiple epitopes. These distinct lineages neutralized a diverse set of HIV-1 viruses from different clades and when combined achieved increased neutralization breadth. With neutralization assays and high-resolution cryo-EM structures, we show that the mAbs used multiple strategies to neutralize the virus: blocking the receptor binding site, binding to HIV-1 Envs N-linked glycans, and disassembly of the trimer.

The distinct antibody lineages neutralized different subsets of viruses suggesting that the serum's remarkable breadth and potency might be the result of a polyclonal mixture rather than a single bNAb lineage. The combined neutralization breadth of the multiple NAbs does not fully recapitulate the serum's breadth suggesting more potent members of the lineages or other bNAb specificities were also present in the serum. While several vaccination strategies focus on a single epitope such as the FP, CD4bs or (V3)-glycan patch [38–41], other immunogens might induce bNAbs targeting several epitope clusters, e.g. the BG505 SOSIP-GT1 trimer [42]. The results here support the rationale for an immunization strategy to induce a polyclonal response against multiple epitopes to protect against infection.

In this study, we have isolated NAbs directed to multiple epitopes including the CD4bs, silent face, gp120-gp41 interface and V3 of HIV-1. The CD4bs-targeting NAbs were derived from the VH1-2*02 germlines and included a normal-length (eight residue) CDRL3, a class of bNAbs that has only been identified once before [43]. In addition, only a few gp120-gp41 interface and silent face bNAbs have been isolated in previous studies [14,15,34]. The reason that only a few of these bNAbs have been isolated so far might be that B cells are generally sorted from HIV-1 infected individuals who have extreme broad and potent neutralizing sera. Most research focuses on the isolation of the most broad and potent bNAbs and therefore might miss bNAbs with lesser neutralization breadth. It will be important to understand if more HIV-1 infected individuals achieve neutralization breadth by a combination of complementary NAbs that target multiple epitopes.

Previously bNAbs have been isolated from HIV-1 infected individuals that target the gp120- gp41 interface, e.g. 3BC315 and 35O22 [31,34]. In addition, an Env-liposome immunogen regimen in rabbits induced bNAb 1C2 that targets a similar region on Env and was able to

neutralize a wide variety of viruses from different subtypes [32]. Despite their close proximity to the viral membrane, these bNAbs can access their epitope and neutralize HIV-1. We identified an additional NAb lineage that binds to a similar epitope as 3BC315 and 1C2. Such antibodies bind between two gp41 subunits, partially disrupting their interface, which in turn might promote trimer dissociation. In contrast, immunization strategies with soluble Env trimers have induced mAbs targeting an overlapping epitope but these are generally non-neutralizing or do so with very limited breadth and potency [44,45], suggesting that display of Env trimers on a viral membrane or liposomes is necessary to elicit bNAbs with an appropriate angle of approach.

Analysis of longitudinal *env* sequences suggests that the lack of the N276 glycan initiated the development of the CD4bs-targeting NAbs in this patient [26]. These findings confirm the rational of using prime immunogens lacking the N276 glycan followed by immunogens harboring the N276 glycan to induce CD4bs bNAbs that can accommodate the N276 glycan. In contrast, the presence of the N295 and N448 glycans was most likely necessary to initiate the antibody lineages in this patient that targeted the silent face of HIV-1 Env, as these glycans were later deleted to escape from neutralization by these lineages. The previously isolated silent face bNAbs VRC-PG05 and SF12 are dependent on the N448 glycan for neutralization, while ACS114 depends on the N295 glycan. In addition, we isolated NAb ACS117 that bound the silent face similarly to VRC-PG05 and SF12, and most likely also depends on the N448 glycan for binding and/or neutralization. The divergence in VH and VL gene segments between the silent face-targeting bNAbs suggests that multiple B cell precursors are present with the potential to develop into silent face bNAbs. However, as VRC-PG05, ACS114 and ACS117 are not very broad, it might be better to include this epitope into a vaccination regimen specifically designed to also induce bNAbs to other epitopes.

Due to the unavailability of *env* sequences between 2- and 8-months post-SC, we could not determine whether silent face and gp120-gp41 interface lineages developed simultaneously or in parallel. However, early members of the silent face lineage contained higher levels of SHM compared those of the gp120-gp41 lineages isolated at a similar timepoint, suggesting that the silent face mAbs arose first. Moreover, the V3-targeting mAbs, ACS130 and ACS131, isolated at 36 months post-SC contained relatively low amount of SHM, suggesting that this lineage arose later during infection. Taken together, the data imply that the first wave of NAbs targeted the CD4bs, followed by the silent face, gp120-gp41 interface and V3-targeting NAbs.

In conclusion, our study reveals that neutralization breadth can be achieved by complementary NAbs targeting various epitopes on HIV-1 Env and supports multi-epitope approaches to achieve this by vaccination.

## Material and methods

### Ethics statement

The Amsterdam Cohort Studies are conducted in accordance with the ethical principles set out in the Declaration of Helsinki, and written consent was obtained prior to data collection. The study was approved by the Academic Medical Center Institutional Medical Ethics Committee.

### Patient samples

PBMCs were obtained from donor H18877 at months 11, 24, 36 and 49 post-SC. Individual H18877 characteristics and serum neutralization have been described previously [5,27]. In brief, donor H18877 was a male participant enrolled in the Amsterdam Cohort Studies on HIV/AIDS who seroconverted during active follow-up. He was infected with a subtype B

HIV-1 variant, did not receive antiretroviral therapy, had detectable viral load, and his CD4 T cell count was stable when samples were collected. In addition, we isolated *env* sequences at various months post-SC from donor 18877. A consensus sequence from *env* sequences of five clonal isolates obtained two months post-SC was generated to produce the AMC009 SOSIP trimer [28].

## Single B cell sorting, RT-PCR, gene amplification and cloning

B cells were isolated from PBMCs obtained at months 2, 11, 24, 36 and 49 post-SC as previously described [46]. Briefly, PBMCs were stained with primary fluorophore-conjugated antibodies to human CD3, CD8, CD14, CD19, CD20, CD27, IgG, IgM (BD Pharmigen) and fluorescently-labeled Env proteins. Avi-tagged AMC009 SOSIP, BG505 SOSIP, 94UG103 gp120 and MGRM C026 gp120 Env proteins were biotinylated and subsequently coupled to Streptavidin-PE (Life Technologies), Streptavidin-APC (Life Technologies), Streptavidin-BV785 (Biolegend) and Streptavidin-PE (Life Technologies), respectively. Unwanted cell populations were first excluded (CD3$^-$/CD8$^-$/CD14$^-$) before HIV-1 Env-specific memory B cells (CD19$^+$/CD20$^+$/IgG$^+$/IgM$^-$) were selected. PBMCs from months 2, 11, 24 and 49 post-SC were sorted with fluorescently labeled AMC009 SOSIP, BG505 SOSIP and 94UG103 gp120 protein, and B cells that could recognize any of the three Env proteins were single cell sorted. In contrast, PBMCs taken at 36 months post-SC were stained with fluorescently labeled 92BR020 gp120, 94UG103 gp120 and BG505 SOSIP protein, and any memory B cell that bound to at least one of the Env proteins was sorted. All staining reactions were performed in PBS supplemented with 1 mM EDTA and 1% FBS for 1 hour at 4˚C. Sorting experiments were performed using a BD FACSAria III machine and cells were collected into 96-well plates containing lysis buffer, and immediately stored at −80˚C.

The mAbs were produced as described previously [33,47,48], i.e., HIV-1 Env-specific B cells were sorted into lysis buffer, mRNA was reverse-transcribed, and polymerase chain reaction (PCR) amplified to produce the HC and LC (kappa or lambda) variable V(D)J regions. PCR reactions were performed using HotStar Taq DNA polymerase master mix (Qiagen) in a total volume of 25 μl volume with 2.5 μl of cDNA transcript and primer sets that have been described previously [47]. Analysis of HC and LC sequences were performed using the IMGT V-quest webserver [49,50]. HC and LC (kappa or lambda) variable regions were cloned into the corresponding Igγ1, Igκ and Igλ expression vectors as previously described [48]. Each antibody lineage was named after the ACS patient identifier (donor H18877, also known as ACS1), followed by the clonal family name (0–3) and a number of the individual clone. For example, ACS101, ACS102, ACS103 are mAbs isolated from HIV-1 infected individual ACS1, lineage 0 with numbers 1, 2 and 3, respectively. Sequences were aligned in BioEdit with a ClustalW Multiple alignment. The phylogenetic tree was generated and visualized with Cipres Science Gateway [51] and FigTree v1.4.4, respectively.

## Monoclonal antibody production and digestion

The mAbs were produced in HEK293F cells as follows (Invitrogen, cat no. R79009). Prior to transfection, HEK293F cells were maintained in FreeStyle Medium (Life Technologies) at a density of ∼1 million cells/ml. Plasmids encoding for the HC and LC were co-transfected at a 1:1 ratio using 1 mg/mL PEImax (Polysciences Europe GmBH, Eppelheim, Germany). Five days after transfection, the HEK293F cell culture was centrifuged and the supernatant was filtered using Steritops (Millipore, Amsterdam, The Netherlands). The mAbs were purified from the supernatant using protein G agarose (Pierce) affinity chromatography columns and eluted with 0.1 M glycine pH 2.5 into 1 M Tris PH 8. Vivaspin20 centrifugal filters with 100 kDa MW

cut-off (Sartorius, Göttingen, Germany) were used to buffer exchange the mAbs into phosphate-buffered saline (PBS).

Monoclonal IgGs were digested to Fab using papain as follows. Papain was activated for 15 minutes at 37°C in a buffer containing 100mM Tris pH 8, 2mM EDTA and 10mM L-cysteine. The mAbs were then incubated with freshly-activated papain for five hours at 37°C and 50mM iodoacetamide was added to stop the mAb digestion by papain. The digestion products were buffer exchanged into tris-buffered saline (TBS) and Fabs were further purified by size exclusion chromatography in TBS. Fractions corresponding to the Fab peak were then pooled and concentrated using an Amicon concentrator.

## AMC009 Env production

The longitudinal *env* sequences were used to produce the SOSIP.v4.2 trimers whereas a consensus sequences of the 2m.1B5, 2m.1C3, 2m.1F9, 2m.1G1 and 2m.2B1 *env* sequences was generated to produce AMC009 SOSIP.v5.2 protein. All AMC009 SOSIP trimers were produced as described previously [52]. In brief, SOSIP trimers were produced in HEK293F cells (Invitrogen, cat no. R79009) by co-transfecting two plasmids encoding the SOSIP trimer and furin using PEImax (Polysciences Europe GmBH, Eppelheim, Germany) as the transfection reagent. Six days after transfection, the HEK293F cell culture was centrifuged and the supernatant was filtered using Steritops (Millipore, Amsterdam, The Netherlands). SOSIP trimers were purified from the supernatant using a PGT145 affinity chromatography column at 4°C as follows. The supernatant was flowed (0.5–1.0 mL/min) over the column followed by wash with PBS and second wash with a buffer containing 20mM TrisHCL pH 8.0, 0.5M NaCl. 3 M $MgCl_2$ pH 7.2, 20 mM TrisHCl was run over the column to elute the SOSIP trimers, which were collected in an equal volume of 20 mM TrisHCl pH 8.0, 75 mM NaCl. Vivaspin20 centrifugal filters with 100 kDa MW cut-off (Sartorius, Göttingen, Germany) were used to concentrate and buffer exchange the mAbs into phosphate-buffered saline (TBS). SOSIP trimers were then further purified by size exclusion chromatography in TBS and fractions corresponding to the SOSIP trimer peak were pooled and further concentrated using an Amicon concentrator.

## Neutralization assays

Neutralization assays with mAbs ACS101-103, ACS110-112, ACS114-115, ACS117, ACS120, ACS122, ACS124-125 and ACS130-131 against viruses from the global panel, the multiclade panel of 119 viruses, and mAb mixtures against the 15-virus panel were performed as follows. Briefly, virus stocks were produced in 293T/17 cells (American Type Culture Collection) by transfection of molecularly cloned Env-pseudotyped virus plasmids. Viruses were titrated in TZM-bl cells prior before use in the assay as described previously [53,54]. Five-fold serial diluted mAbs (start concentration of 50 μg/ml) were incubated with a pre-titrated dose of virus in a total volume of 150 μl in 96-well flat-bottom culture plates for 1 hour at 37°C. The mAb mixtures were tested such that each mAb in the mixture was at a primary concentration of 50 μg/ml. Freshly trypsinized ~10,000 cells TZM-bl cells (obtained from the NIH AIDS Research and Reference Reagent Program, as contributed by John Kappes and Xiaoyun Wu, also called JC57BL-13) were added to each well in 100 μl of growth medium containing DEAE-Dextran and incubated for 48 hours at 37°C. The TZM-bl cell line is a HeLa cell clone engineered to express CD4 and CCR5 [55] and contains integrated reporter genes for firefly luciferase and *E. coli* beta-galactosidase under control of an HIV-1 LTR [56]. Wells with virus or TZM-bl cells only were taken along as a positive and negative control, respectively. After incubation, luminescence was measured using the Britelite Luminescence Reporter Gene Assay System (PerkinElmer Life Sciences). Neutralization $IC_{50}$ was determined as the dilution

at which relative luminescence units were reduced by 50% compared to the virus only controls minus the background of the TZM-bl cells only control. The assay was performed in a laboratory in compliance with Good Clinical Laboratory Practices, including participation in a formal proficiency testing program [57], and has been previously optimized and validated [58]. All supporting protocols may be found at: http://www.hiv.lanl.gov/content/nab-reference-strains/html/home.htm

Neutralization assays of mAbs ACS113, ACS116, ACS121, ACS123 and ACS126 against the global panel viruses and ACS130-131 and ACS114 against the JRCSF mutant panels were performed as previously described [53,59]. In brief, viruses were produced in HEK293T cells (ATCC, CRL-11268) by transfecting the molecularly cloned Env-pseudotyped virus plasmids using lipofectamin2000 (Invitrogen). After three days, the viruses were harvested and stored at -80˚C. Half-area 96-wells plates were seeded with ~17.000 luciferase reporter TZM-bl cells (obtained through the NIH AIDS Reagent Program, Division of AIDS, NIAID, NIH from Dr. John C. Kappes, and Dr. Xiaoyun Wu). The mAbs (start concentration 50 µg/ml) were three-fold serial diluted and incubated with titrated Env-pseudotyped viruses for one hour at room temperature (RT) in duplicate. Wells with virus or TZMbl-cells only were taken along as a positive and negative control, respectively. After one hour, DEAE (40 µg/ml) and sanquinvir (400 nM) were added to the TZM-bl-cells followed by the mAb/virus mixture and incubated for three days at 37˚C. The Bright-Glo (Promega) and GloMax Discover System was used to measure luciferase activity and data was analyzed using GraphPad Prism version 9.0.

## Bio-layer interferometry with longitudinal AMC009 SOSIP proteins

Bio-layer interferometry experiments were performed at RT using an Octet Red96 instrument (ForteBio). Anti-human IgG (AHI) probes were equilibrated in reaction buffer (PBS pH 7.4 + 0.01% (w/v) BSA + 0.002% (v/v) Tween 20) for 60 seconds. The mAbs were immobilized onto the AHI probes at 10 µg/ml in reaction buffer for 300 seconds. The probes were then washed for 60 seconds in reaction buffer, and binding to SOSIPv4.2 trimers at a concentration of 250 nM was measured for 300 seconds. Dissociation of the SOSIPv4.2 trimers was measured for 300 seconds in reaction buffer. Analyses were performed using Octet software and GraphPad Prism version 9.0.

## Bio-layer interferometry for antibody binding kinetics

To determine antibody binding kinetics, His-tagged antigen (AMC009 SOSIP protein) was immobilized on Ni-NTA biosensor (ForteBio) at 10 ug/mL with diluted mAbs (IgG) used as analytes. Both antigen and mAbs were diluted in reaction buffer (1x PBS, 0.1% BSA, 0.002% Tween). Briefly, hydration of Ni-NTA biosensor in the reaction buffer 10 min prior to the start of the assay, followed by 3 cycles of regeneration (buffer, 1X PBS 10 mM glycine), neutralization (buffer, 1x PBS, 0.1% BSA, 0.002% Tween) and activation (buffer, 1x PBS, 1 µM $NiCl_2$) at 15 seconds per step followed by a baseline step (1x PBS, 0.1% BSA, 0.002% Tween) at 180 seconds of baseline. AMC009-His at 10 µg/ml in 200 µL reaction buffer were loaded on the Ni-NTA biosensor for 300 seconds followed by 60 seconds of baseline before association of the mAbs for 600 seconds and subsequently 600 seconds of dissociation. The method above was repeated for each mAb and mAb dilutions. All steps were performed at 1000 rpm using Octet Red instrument (ForteBio).

## Bio-layer interferometry for antibody synergy assay

To determine antibody synergies against AMC009, a modified version of antibody binding kinetics assay was performed. Briefly, 10 µg/mL of AMC009 was immobilized in 200 uL of

reaction buffer (1x PBS, 0.1% BSA, 0.002% Tween) for 600 seconds using ForteBio's Ni-NTA biosensor followed by 60 seconds of baseline (1x PBS, 0.1% BSA, 0.002% Tween). Subsequently, antibodies were associated to AMC009-loaded biosensor at 10 µg/ml in reaction buffer for 300 seconds one antibody at a time as listed in the figure without the subsequent antibody dissociation step. All steps were performed at 1000 rpm using Octet Red instrument (ForteBio)

## Negative stain electron microscopy

Fab/IgG-SOSIP complexes were made by incubating Fabs or IgGs with AMC009 SOSIP trimers with a 4-fold and 2-fold molar excess of antibody, respectively, for 30 minutes at RT. The Fab/IgG-SOSIP complexes were loaded onto glow-discharged, carbon-coated Cu400 EM grids at a concentration of 30 ng/µl in TBS for 10 seconds. The grids were blotted to remove excess sample and the Fab/IgG-SOSIP complexes were then stained with 2% (w/v) uranyl formate. Grids were immediately blotted and a second stain with 2% (w/v) uranyl formate was applied for 30 seconds, followed by a final blot to remove excess stain. A Tecnai Spirit T12 (FEI) (120kV, 52,000x magnification) equipped with an Eagle 4K CCD (FEI/Thermo Fisher) or Tecnai T20 (FEI) (200kV, 62,000x magnification) equipped with a TemCam F416 CMOS (TVIPS) was used to image the grids. Image collection was performed using Leginon and data processing was carried out as previously described [60,61]. 2D classification and 3D sorting was performed with Relion v3.0 [62], and UCSF Chimera [63] and Segger [64] were used to visualize and segment the EM maps, respectively.

## Cryo-electron microscopy data collection

300 µg of AMC009 SOSIPv5.2 trimer was incubated with 10-fold molar excess of ACS114 Fab for 12 hours at RT. A 6-fold molar excess of ACS122 Fab was added for 15 minutes before the ACS122-ACS114-AMC009-SOSIP complex was purified by size exclusion chromatography in TBS. Fractions corresponding to the ACS122-ACS114-AMC009-SOSIP complex were pooled and concentrated to 7.0 mg/ml using an Amicon concentrator (Millipore). Cryo-EM grids were prepared using a Vitrobot mark IV (Thermo Fisher Scientific) with the humidity set to 100% and a chamber temperature of 10˚C. Blotting time was varied within a 6.0–7.5 seconds range and wait time was set to 10 seconds. No blot force was applied to the cryo-EM grids. Quantifoil (R 1.2/1.3, 400) grids were treated with Ar/O$_2$ (Solarus plasma cleaner, Gatan) for 10 seconds before sample application. For cryo-EM grid preparation, the ACS122-ACS114-AMC009-SOSIP complex was mixed with either n-Dodecyl-β-D-Maltopyranoside (DDM) or lauryl maltose neopentyl glycol (LMNG) at a final concentration of 0.06 and 0.005 mM, respectively, and 3 µl was immediately loaded onto the grid. After the blot step, the grids were plunge-frozen into liquid ethane. Data were collected on a FEI Talos Arctica electron microscope (ThermoFisher) operating at 200 keV using Leginon software [60]. A K2 Summit direct electron director camera (Gatan) was set to 36,000 with the resulting pixel size at the specimen plane of 1.15 Å.

## Cryo-electron microscopy data processing

Micrograph movie frames were aligned and dose-weighted with MotionCor2 [65] and GCTF [66] was used to estimate CTF parameters. All further processing was performed in cryoSPARCv3.2.0 [67]. Additional information regarding data collection and processing of the ACS122-ACS114-AMC009-SOSIP complex can be found in S1 Table. SAbPred [68] was used to generate initial Fab models. These initial Fab models were then docked together with the Env-portion of PDB: 6VO3 into the EM map of the ACS122-ACS114-AMC009-SOSIP

complex using UCSF Chimera [63]. Coot [69,70] was used add N-linked glycans [71] and to introduce the additional mutations for SOSIPv5.2 compared to SOSIPv4.2 (PDB: 6V03). The models were refined into the EM map with iterative rounds of manual model building in Coot and RosettaRelax [72,73]. Evaluation of the final models was performed with MolProbity [74] and EMringer [75]. Buried surface area calculations and distance measurements were performed using PDBePISA [76] and predicted hydrogen bonds and salt bridges were assigned with a distance <3.2 Å.

## Protein production, and crystallization

For crystallography, Expi293F cells were transiently transfected with ACS122 Fab HC and LC plasmids. ACS122 Fabs were purified using CH1-XL affinity matrix beads (Thermo Scientific) and size exclusion chromatography using a Superdex75 column (GE Healthcare). Our automated CrystalMation robotic system (Rigaku) was used to set up crystallization trials of ACS122 Fab at a concentration of 9.5 mg/ml at Scripps Research. Crystals of ACS122 Fab were obtained at 4˚C in a solution at 0.1M of Tris (pH 8.5) and 55%(v/v) 2-methyl-2,4-pentanediol. Crystals of ACS122 Fab were cryo-protected with reservoir crystallization solution followed by fast plunging into liquid nitrogen and storing before data collection.

## X-ray data collection, data processing and structure determination

Diffraction data from crystals of ACS122 Fab were collected at Stanford Synchrotron Radiation Lightsource beamline 12–1. HKL2000 [77] was used for data processing. The coordinates from germline precursor of 3BNC60 Fab [78] (PDB ID: 5F7E, 1.90Å) with CDR loops removed were used as a search model for molecular replacement and refinement in PHENIX [79]. All the model building was performed in Coot [80]. Additional information on the data collection and refinement statistics can be found in S2 Table.

## Supporting information

**S1 Fig. Sequence characteristics of the various antibody lineages targeting four different epitopes.** Somatic hypermutation (SHM) is determined as the percentage of nucleotide differences with the VH, VL and VK gene segments.
(TIF)

**S2 Fig.** Antibody heavy and light chain sequence comparisons to the VH, VL or VK germline gene segments for mAbs targeting the (a) CD4-binding site, (b) silent face, (c) gp120-gp41 interface and (d) V1/V1/V3 region.
(TIF)

**S3 Fig. Neutralization of nine viruses from the global panel and the autologous AMC009 virus.** (**a**) Neutralization potency (antibody concentration (μg/ml) that inhibits 50% of viral infectivity ($IC_{50}$) (μg/ml) is presented in table format. The mAbs were tested at a starting concentration of 50 μg/ml. (**b**) Neutralization potency (antibody concentration (μg/ml) that inhibits 50% of viral infectivity ($IC_{50}$)) is depicted as a dot plot. The tested viruses are indicated along the horizontal axis. Each symbol represents a mAb with the different colors indicating the epitope targeted. We also provide the estimated neutralization breadth if the most potent NAbs were combined and tested against the global panel viruses.
(TIF)

**S4 Fig. Neutralization of a multiclade 119-virus panel by mAbs by ACS101, ACS114, ACS117, ACS122, ACS125 and ACS130.** Breadth is indicated below as the percentage of

viruses that were neutralized (upper panel). Geometric mean of the $IC_{50}$ (μg/ml) is given as the antibody concentration that inhibits 50% of viral infectivity (lower panel). We also provide an estimated neutralization breadth/$IC_{50}$ of all mAbs combined.
(TIF)

**S5 Fig.** Combined binding and neutralization of antibody lineages (a) Simultaneous binding of ACS101, ACS114, ACS122 and ACS130 to AMC009 SOSIP trimer using bio-layer interferometry. (b) Neutralization of a multiclade 15-virus panel by ACS101, ACS114, ACS117, ACS122 and ACS130 individually or combined. The virus panel was based on positive hits from the large (n = 119) multiclade virus panel. Neutralization potency (antibody concentration (μg/ml) that inhibits 50% of viral infectivity (IC50) is given.
(TIF)

**S6 Fig. Cryo-EM and X-ray data collection.** (**a**) Cryo-EM map of ACS122 and ACS114 in complex with the AMC009 SOSIP trimer. Only the Fab variable regions are shown here. (**b**) Cryo-EM map of ACS122 and ACS114 in complex with AMC009 SOSIP colored by local resolution (Å). (**c**) Gold-standard Fourier shell correlation (GSFSC) resolution estimate of ACS122 and ACS114 Fabs in complex with AMC009 SOSIP. (**d**) Crystal structure of unliganded ACS122 Fab (HC; purple and LC; light pink) at 1.84 Å resolution.
(TIF)

**S7 Fig. ACS114 depends on the N295 glycan for neutralization.** (**a**) Neutralization potency (antibody concentration (μg/ml) that inhibits 50% of viral infectivity ($IC_{50}$) of ACS114 against a panel of JRCSF mutants. The different mutants are indicated along the horizontal axis. The mAbs were tested at a starting concentration of 50 μg/ml. (**b**) The NS-EM map of ACS117 with the structure of VRC-PG05 (PDB:6BF4) and SF12 (PDB:6OKQ) docked in. (**c**) The NS-EM map of ACS114 with the structure of VRC-PG05 (PDB:6BF4) and SF12 (PDB:6OKQ) docked in.
(TIF)

**S8 Fig. Interactions of ACS122 with the N88 glycan and gp41.** (**a**) Interactions of ACS122 with the N88$_{gp120}$ glycan (left), FPPP (middle) and the tryptophan clasp and surroundings (right). The N88 glycan is shown as sticks and contoured by the cryo-EM map at 1σ. Amino acid interactions between ACS114 and gp120 are highlighted based on the density in the cryo-EM map and predicted hydrogen bonds are shown with a distance <3.2 Å. (**b**) Comparison of ACS122 to 1C2 (PDB:6PEH). Fabs and trimer are shown as a surface representation and the magnified view is depicted as ribbons. The structures were aligned relative to gp41.
(TIF)

**S9 Fig. Binding of mAbs to longitudinal Env using bio-layer interferometry (BLI).** Binding curves are shown for mAbs targeting the (**a**) CD4-binding site, (**b**) silent face, (**c**) gp120-gp41 interface, (**d**) V3 region and (**e**) control bNAbs. Binding curves for different longitudinal Envs are colored as indicated on the right.
(TIF)

**S10 Fig. Binding of mAbs to AMC009 SOSIP using bio-layer interferometry (BLI).** Binding curves are shown for various mAbs targeting the (**a**) CD4-binding site, (**b**) silent face, (**c**) gp120-gp41 interface, (**d**) V3 region. (**e**) Overview of the KD's of the tested mAbs. Binding curves for mAbs with different concentrations are colored as indicated on the right.
(TIF)

**S1 Table. EM data collection and map/model refinement parameters.**
(TIF)

**S2 Table. X-ray data collection and refinement statistics.** Values in parentheses correspond to the highest resolution shells.
(TIF)

**S3 Table. BLI binding kinetics.**
(TIF)

## Acknowledgments

The Amsterdam Cohort Studies (ACS) on HIV infection and AIDS, a collaboration between the Amsterdam Health Service, the Academic Medical Center of the University of Amsterdam, Sanquin Blood Supply Foundation, and the Jan van Goyen Clinic, are part of The Netherlands HIV Monitoring Foundation. We thank Mitch Brinkkemper for providing SOSIP trimers used in the bio-layer interferometry experiments.

## Author Contributions

**Conceptualization:** Jelle van Schooten, Rogier W. Sanders, Andrew B. Ward, Marit J. van Gils.

**Data curation:** Jelle van Schooten, Gabriel Ozorowski.

**Formal analysis:** Jelle van Schooten, Elinaz Farokhi, Gabriel Ozorowski, Marit J. van Gils.

**Funding acquisition:** David C. Montefiori, Dennis R. Burton, Michael S. Seaman, Ian A. Wilson, Rogier W. Sanders, Andrew B. Ward, Marit J. van Gils.

**Investigation:** Jelle van Schooten, Anna Schorcht, Elinaz Farokhi, Jeffrey C. Umotoy, Hongmei Gao, Tom L. G. M. van den Kerkhof, Jessica Dorning, Tim G. Rijkhold Meesters, Patricia van der Woude, Judith A. Burger, Tom Bijl, Riham Ghalaiyini, Alba Torrents de la Peña, Hannah L. Turner, Celia C. Labranche, Robyn L. Stanfield, Devin Sok, Gabriel Ozorowski, Marit J. van Gils.

**Methodology:** Jelle van Schooten.

**Resources:** Hanneke Schuitemaker, David C. Montefiori, Dennis R. Burton, Michael S. Seaman, Ian A. Wilson, Rogier W. Sanders, Andrew B. Ward.

**Supervision:** Marit J. van Gils.

**Validation:** Jelle van Schooten, Gabriel Ozorowski.

**Visualization:** Jelle van Schooten.

**Writing – original draft:** Jelle van Schooten, Rogier W. Sanders, Marit J. van Gils.

**Writing – review & editing:** Jelle van Schooten, Dennis R. Burton, Ian A. Wilson, Rogier W. Sanders, Marit J. van Gils.

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
