## [Decision Letter · Decision Letter 0]

6 Jun 2022

Dear Mr. van Schooten,

Thank you very much for submitting your manuscript "Complementary antibody lineages achieve neutralization breadth in an HIV-1 infected elite neutralizer" for consideration at PLOS Pathogens. As with all papers reviewed by the journal, your manuscript was reviewed by members of the editorial board and by several independent reviewers. The reviewers appreciated the attention to an important topic. Based on the reviews, we are likely to accept this manuscript for publication, providing that you modify the manuscript according to the review recommendations.

Sincerely,

Katie J Doores

Associate Editor

PLOS Pathogens

Susan Ross

Section Editor

PLOS Pathogens

Kasturi Haldar

Editor-in-Chief

PLOS Pathogens

orcid.org/0000-0001-5065-158X

Michael Malim

Editor-in-Chief

PLOS Pathogens

orcid.org/0000-0002-7699-2064

Reviewer Comments (if any, and for reference):

Reviewer's Responses to Questions

**Part I - Summary**

Reviewer #1: Schooten et al

To the editor

To the authors

The authors describe broadly neutralizing MAb isolation from an elite neutralizing HIV+ donor. They describe multiple bnAbs with various specificities that collectively do not fully recapitulate the breadth of the donor serum, and conclude that inducing combinations of moderate bnAbs could be a viable vaccine strategy.

The paper is another high capacity mAb recovery and detailed characterization study, which is by now a well-beaten path that this group and others have perfected in recent years. The authors then show that viral escape in the donor is driven by the two most potent lineages (CD4 binding site and silent face). Overall, the paper is well-written and succinct. A few minor comments and suggestions follow.

ACS130: the epitope of some mabs is called V1V2V3, but further studies suggest that it is V3 directed, so this may be a “straw man” for review? It seems possibly slightly misleading to say it is V1V2V3 in the earlier parts of the paper including the abstract as this gives the impression of a new complex V1V2V3 epitope or possibly V1V2 apex epitope. Is the neutralizing activity adsorbed by V3 peptides?

Line 110, 152, 220. To say “we previously described…” then to cite a paper that is not published seems contradictory unless this other paper is published while this one is in review.

Line: 139-140. I am not sure if it is OK to say ‘data not shown’ for this journal?

The interface mAbs show weak neutralizing titers >10ug/ml, compared to some of the known ones. These ones also exhibit the least breadth, although the breadth and potency of the other Mabs is not particularly high, hence the suggestion that they be used together. Do the authors know what might make these MAbs less potent than some of the published bnAbs? For example, do multiple lineages mean there is less successful SHM in each as compared to when fewer lineages are developing? Or are these mabs simply less potent versions of other mabs in the donor that have not been mined out of the donor yet? Is it true that broadly neutralizing donors always harbor lineages with a range of breadth and potency, including weaker ones, as compared to a few potent and broad mabs, as prior studies imply?

Line 253. Why would a 73-561 disulfide mitigate mab induced protomer dissociation? A lateral disulfide might prevent it, though. Does the interface mab cause dissociation of the real virus trimers?

Line 309. The text talks about mutations in the epitope of the interface mAb, but the mAb binds equally to them. Nevertheless, the authors say that the mutations indicate when the mAb was generated at 2-8 months. However, since the mutations, although close to the epitope don’t impact binding per se, I don’t follow the reasoning, as no “escape” seems to be

Reviewer #2: The manuscript by van Schooten and colleagues identified and characterized three complementary bNAb lineages from an elite neutralizer, which targeted non-overlapping Env epitopes, including the silent face, gp120-gp41 interface, and V1/V3 region. The authors performed comprehensive biochemical and structural analyses of longitudinal samples to provide the basis of antibody neutralization and env evolution. While no single antibody lineage recapitulated the complete plasma neutralizing activity, the polyclonal mixture of select antibodies from these three lineages (plus a CD4bs lineage identified from the same donor and described in a related manuscript) was able to yield potent and broad neutralizing activity. Thus, the authors postulate that vaccine strategies capable of eliciting multiple antibody lineages that target complementary Env epitopes with moderate breadth and potency is an achievable goal. While the studies in this manuscript are well carried out and written, there are some points the authors should address in a revised manuscript prior to publication.

**Part II – Major Issues: Key Experiments Required for Acceptance**

Reviewer #1: (No Response)

Reviewer #2: The authors performed neutralization studies of antibodies, but missed an opportunity to provide critical data on the effect of affinity to Env epitopes for antibodies within the same lineage isolated at different timepoints. For example how does ACS110 affinity compare to ACS114? Additionally, at timepoints 2 and 8 mo seroconversion the authors suggest in lines 299-300 and 314 that this is when the silent face and gp41/gp120 lineages developed. Did the authors observe intermediate antibodies with less somatic mutation at these timepoints for those lineages? If so, how do the affinities of these antibodies compare to the mature neutralizing antibodies? What might this say about the level of SHM necessary to elicit neutralizing antibodies? Additionally, the authors show that when combined, there is no synergy between the different antibody classes. However, outside of the ACS114-ACS122-Env structure, the authors do not show if the antibodies can target Env simultaneously. The authors should provide competition BLI or ELISA binding data, which may point at the inability of antibodies to synergize.

Given the finding that ACS122 binds Env in a similar way to 1C2 and also disrupts the tryptophan clasp, it’s surprising that the ACS122 doesn’t show broad neutralizing activity. Antibody 1C2 was capable of moderately neutralizing 87% of isolates in a 208-virus panel. A more thorough comparison of ACS122 and 1C2 would be appreciated, particularly, what are the major differences that give rise to broad neutralization for 1C2, but limit ACS122?

**Part III – Minor Issues: Editorial and Data Presentation Modifications**

Reviewer #1: (No Response)

Reviewer #2: For silent face antibodies, removal of glycans at position 295 or 448 disrupted binding. However, env sequences in the C4 region included addition of a glycan at position 444, which would appear to clash with ACS114 binding. However, BLI data against Env sequence 27m.2C4 shows binding in the presence of the N442-glycan. Could the authors speculate how ACS114, in particular, can accommodate this glycan?

Line 101: “..published results of the AMP trials implies..” should be “imply.”

Line 162: “In addition, we isolated an eight member..” should be “eighth member.”

Line 218: “.. HIV-1 vaccine design how to induce similar..”, should remove “how”

Line 250: “Furthermore, ACS112 CDRH1..”, should this be ACS122?

PLOS authors have the option to publish the peer review history of their article (what does this mean?). If published, this will include your full peer review and any attached files.

Reviewer #1: No

Reviewer #2: No

Figure Files:

Data Requirements:

Reproducibility:

References:

---

## [Editor Report · Decision Letter 1]

21 Oct 2022

Dear Mr. van Schooten,

We are pleased to inform you that your manuscript 'Complementary antibody lineages achieve neutralization breadth in an HIV-1 infected elite neutralizer' has been provisionally accepted for publication in PLOS Pathogens.

Best regards,

Katie J Doores

Associate Editor

PLOS Pathogens

Susan Ross

Section Editor

PLOS Pathogens

Kasturi Haldar

Editor-in-Chief

PLOS Pathogens

orcid.org/0000-0001-5065-158X

Michael Malim

Editor-in-Chief

PLOS Pathogens

orcid.org/0000-0002-7699-2064
---

## [Editor Report · Acceptance letter]

11 Nov 2022

Dear Dr. van Gils,

We are delighted to inform you that your manuscript, "Complementary antibody lineages achieve neutralization breadth in an HIV-1 infected elite neutralizer," has been formally accepted for publication in PLOS Pathogens.

Best regards,

Kasturi Haldar

Editor-in-Chief

PLOS Pathogens

orcid.org/0000-0001-5065-158X

Michael Malim

Editor-in-Chief

PLOS Pathogens

orcid.org/0000-0002-7699-2064